# Experimental Procedure for Fifth Generation (5G) Electromagnetic Field (EMF) Measurement and Maximum Power Extrapolation for Human Exposure Assessment

**Daniele Franci [1,\*], Stefano Coltellacci [1], Enrico Grillo [1], Settimio Pavoncello [1], Tommaso Aureli [1], Rossana Cintoli [1] and Marco Donald Migliore [2,3]**

[1] ARPA Lazio (Agenzia per la Protezione Ambientale del Lazio), Via G. Saredo 52, 00172, Rome, Italy; stefano.coltellacci@arpalazio.gov.it (S.C.); enrico.grillo@arpalazio.gov.it (E.G.); settimio.pavoncello@arpalazio.gov.it (S.P.); tommaso.aureli@arpalazio.gov.it (T.A.); rossana.cintoli@arpalazio.it (R.C.)

[2] DIEI (Dipartimento di Ingegeria Elettrica e dell'Informazione "Maurizio Scarano")—University of Cassino and Southern Lazio, via G. Di Biasio 43,03043 Cassino, Italy; mdmiglio@unicas.it

[3] ICEmB(Inter-University National Research Center on Interactions Between Electromagnetic Fields and Biosystems), Via All'Opera Pia, 11 A, 16145 Genova GE, Italy

\* Correspondence: daniele.franci@arpalazio.gov.it

**Abstract:** The fifth generation (5G) technology has been conceived to cover multiple usage scenarios from enhanced mobile broadband to ultra-reliable low-latency communications (URLLC) to massive machine type communications. However, the implementation of this new technology is causing increasing concern over the possible impact on health and safety arising from exposure to electromagnetic field radiated by 5G systems, making imperative the development of accurate electromagnetic field (EMF) measurement techniques and protocols. Measurement techniques used to assess the compliance with EMF exposure limits are object to international regulation. The basic principle of the assessment is to measure the power received from a constant radio frequency source, typically a pilot signal, and to apply a proper extrapolation factor. This kind of approach is standardized for 2G, 3G, and 4G technologies, but is still under investigation for 5G technology. Indeed, the use of flexible numerologies and advanced Time Division Duplexing (TDD) and spatial multiplexing techniques, such as beam sweeping and Massive Multiple Input Multiple Output (MIMO), requires the definition of new procedures and protocols for EMF measurement of 5G signals. In this paper a procedure for an accurate estimation of the instant maximum power received from a 5G source is proposed. The extrapolation technique is based on the introduction of proper factors that take into account the effect of the TDD and of the sweep beam in the measured value of the 5G signal level. Preliminary experimental investigation, based on code domain measurement of appropriate broadcast channels, and carried out in a controlled environment are reported, confirming the effectiveness of the proposed approach.

**Keywords:** 5G NR; TDD; beam sweeping; SS-Block; Massive MIMO; EMF exposure

## 1. Introduction

The fifth generation (5G) technology radio interface, known as New Radio (NR) [1–3], represents a dramatic step forward in wireless technology compared to the previous generations. Moreover, 5G technology has been conceived to cover multiple usage scenarios, from enhanced mobile broadband

to ultra-reliable low-latency communications (URLLC), to massive machine type communications. From a technical point of view, this required the exploitation of new paradigms, like the use of millimeter waves, variable numerology, bandwidth parts, and a more sophisticated implementation of techniques introduced in 4G, such as Time Division Duplex (TDD) and Multi-User Multiple Input Multiple Output (MU-MIMO) [4] techniques. NR takes advantage of a high degrees of flexibility at the code domain level (e.g. variable numerology), at the time domain level (e.g. variable TDD schemes), at the frequency domain level (e.g. bandwidth parts) and at the spatial domain level (e.g. high flexibility in the implementation of beam sweeping or MU-MIMO technology) to optimize the use of temporal and spatial resources given by the communication channel.

While the advantages of 5G are well discussed, and there is no doubt about the necessity of deployment of faster and more reliable wireless communication systems, on the other hand, the implementation of this new technology is causing increasing concern over the possible impact on health and safety arising from exposure to the electromagnetic field radiated by 5G systems—making imperative the development of accurate EMF measurement techniques and protocols.

Measurement techniques used to assess the compliance with electromagnetic field (EMF) exposure limits are the object of international regulation. In particular, the International Electrotechnical Commission (IEC) has recently published IEC 66232 "Determination of RF field strength, power density and SAR in the vicinity of radiocommunication base stations for the purpose of evaluating human exposure" and IEC Technical Report (TR) 62669 "Case studies supporting IEC 66232" [5,6]. This standard addresses the evaluation of radio frequency (RF) field strength, power density, and Specific Absorption Rate (SAR) levels in the vicinity of radiocommunication base station (RBS) radiating in the frequency range 110 MHz to 100 GHz. The basic principle of the assessment is to measure the power received from a constant radio frequency source, typically a pilot signal, and to apply a proper extrapolation factor as described in [5] annex B.5. This method ensures that the resulting field is the maximum obtainable at the location for the considered radiofrequency source. This kind of approach is standardized for 2G, 3G, and 4G technologies as described in Appendix F, "Technology-specific guidance" [5] but is still under investigation for 5G technology. Indeed, the use of flexible numerologies and advanced TDD and spatial multiplexing techniques, such as beam sweeping and Massive MIMO, requires the definition of new procedures and protocols for EMF measurement of 5G signals.

Extrapolation technique is of great practical relevance. Indeed, exposure limits of the population refer to measured quantities averaged over a period of time. In order to reduce the costs of measurements, it is possible to follow a different procedure that requires to multiply the instant maximum power by a proper scaling factor. In order to apply such a procedure, the development of effective extrapolation techniques, able to estimate the instant maximum power from measurements carried out in a short period of time, is of paramount importance.

In this paper an extrapolation procedure for an accurate estimation of the instant maximum power received from a 5G source is proposed. The procedure is based on the choice of an effective pilot channel, whose received power is eligible to be the reference for the extrapolation technique. The extrapolation formula includes proper factors that take into account the effect of the TDD and of the sweep beam in the measured value of the 5G signal level. A specific experimental set-up was built in laboratory in order to validate the proposed extrapolation technique. The results of the experimental investigation, reported in this paper, confirm the effectiveness of the technique. The use of a specifically designed experimental set-up was motivated to overcome two main problems. The first is related to the very small number of 5G stations currently implemented in Italy. In fact, the implementation of 5G is still at the first stage, and the few active base stations are mainly used for tests that allow very limited access to the on-air signals. The second is the requirement of stable and repeatable conditions necessary to verify the methods. The solution to the above described problems were obtained in two stages. In the first phase, a set of signals transmitted during a test of a 5G base station were measured on-air. The characteristics of the signals are described in [7]. Then a laboratory set-up was created capable of reproducing the acquired signals. This approach has several advantages. In particular, it allows to work with an exact copy of real 5G signals in laboratory, whose

parameters are completely known. Furthermore, it is possible to vary some signal parameters to consider different cases. For example, in the experimental examples reported in this article, the proposed extrapolation technique is applied to signals having an increasing 'degree of complexity'. It is important to stress again that the signals are copies of real 5G signals.

As further observation, it must be noted that the problem of EMF extrapolation procedures has a great practical impact in 5G deployment; hence, has attracted the attention of other research groups [8,9]. However, this is a new field of research, and first results have been available in the open literature only very recently.

Finally, the work described in this paper is closely related to the research described in [7] and represents its natural continuation. In [7], the procedures to estimate $\langle E_{RE}^{DMRS-PBCH} \rangle$ and $F_{TDC}$ are introduced. In this paper we proposed an extrapolation technique that uses the parameters introduced in [7], in addition to other parameters. Consequently, this article only gives a brief reference to the relationship between Synchronization Signal (SS) Burst power and switched beam strategy used for 5G SS Blocks is reported in this paper. The reader is invited to refer to [7] for further information on aspects concerning the above-mentioned two parameters.

## 2. EMF Level Estimation Using Extrapolation Techniques in Previous Cellular Systems

The radiated power of modern cellular communications systems varies with time, depending on a number of factors, including data traffic variation and dynamic power control used in the communication link. A simple and effective approach to take into account these factors is to use extrapolation techniques.

Broadly speaking, estimation of the average or of the maximum value of a time variant signal requires a component of the signal transmitted at constant power level, which is used as reference.

In the Global System for Mobile (GSM) communication base station, the Broadcast Control Channel (BCCH) signal is always broadcasted with constant and maximum power. Consequently, the EMF level (e.g. the measured electric field magnitude) of the BCCH ($E_{BCCH}$) is an excellent candidate for EMF level estimation. The maximum EMF level in a specific GSM cell sector is determined as

$$E_{2G}^{max} = \sqrt{\alpha}\, E_{BCCH} \tag{1}$$

where in $\alpha$ is the number of carrier frequencies (BCCH plus traffic carriers) in one cell sector. The level of BCCH signal can be measured in the frequency domain using a scalar spectrum analyzer.

In case of the Universal Mobile Telecommunications System (UMTS), the same procedure requires a more complex instrument. UMTS extrapolation technique uses the P-CPICH (Primary-Common Pilot Channel), which is always broadcasted in each cell sector of the UMTS network, as reference signal. The maximum EMF level is estimated as

$$E_{3G}^{max} = \sqrt{R_{P-CPICH}}\, E_{P-CPICH} \tag{2}$$

where in $E_{P-CPICH}$ is the P-CPICH measured electric field magnitude and $R_{P-CPICH}$ is the extrapolation factor as described in the IEC 62232:2017 standard [5]. The P-CPICH ($E_{P-CPICH}$) is encoded in the 3G signal, and its measurement requires the decoding of the signal using a Vector Spectrum Analyzer (VSA).

The same approach is also applied to the Long Term Evolution (LTE) signals. In LTE the reference level is obtained from the Cell-Specific Reference Signal (CRS). The CRS is transmitted in the subframes of the Physical Downlink Shared Channel (PDSCH), and its level ($E_{CRS}$) can be obtained by measurements in the code domain using a VSA. The maximum EMF level is estimated, applying a proper extrapolation factor $K_{CRS}$ as

$$E_{4G}^{max} = \sqrt{K_{CRS}}\, E_{CRS} \tag{3}$$

This short review of the extrapolation techniques from 2G to 4G clearly show the critical role played by the reference signals. Loosely speaking, identification of the reference signal, as well as evaluation of the extrapolation factor, becomes more complex with the introduction of a new

generation of cellular communication systems, forcing the use of a VSA. In any case, a common characteristic in the previous generation of cellular systems is the constant received power level of the reference signal.

## 3. Brief Introduction to 5G Technology and Comparison with 4G

In this section, a brief technical introduction to the major similarities and differences between 4G and 5G systems is presented, focusing on the impact that these features have on the definition of an effective extrapolation technique for the estimation of the instant maximum power received at a given point from a 5G source. A detailed analysis of 5G signal is outside the scope of this paper. The interested reader can find details in [10]. A short review of the main characteristic that have an impact on EMF measurements is also reported in [7].

### 3.1. Some General Characteristics of 5G VS. 4G

While 4G uses only frequencies lower than 6 GHz, NR supports two bandwidths: Frequency Range 1 (FR1), commonly referred to as sub-6 GHz, which ranges from 450 MHz to 7125 GHz, and Frequency Range 2 (FR2), commonly referred to as millimeter wave, which ranges from 24 GHz to up to 50 GHz. The maximum bandwidth in FR1 is 100 MHz, while the maximum bandwidth in FR2 is 400 MHz. These values are much larger compare to the LTE bandwidth, limited to 20 MHz. Both 4G and 5G use Orthogonal Frequency Division Multiplexing (OFDM). However, while in LTE the subcarrier spacing is fixed and equal to 15 KHz, NR introduces a flexible numerology that allows the use of variable subcarrier spacing. In particular, the subcarrier spacing can be chosen equal to $2^\mu \cdot 15$ kHz, with $\mu = 0, 1, 2, 3, 4.$, is scaled up by a factor of $2^\mu$ from the LTE subcarrier space. Modulation schemes are similar to LTE and include Binary Phase Shift Keying (BPSK), Quadrature Phase Shift Keying (QPSK), Quadrature Amplitude Modulation of order 16 (16 QAM), 64 QAM, and 256 QAM. The time length of both the LTE and NR frame is equal to 10 ms and consists of 10 subframes, each of them having a time length of 1 ms. However, in the NR different numerologies give a different number of OFDM symbols in a subframe. Accordingly, in NR, the subframe is divided in the $2^\mu$ slot of 14 OFDM symbols each (12 symbols for extended Cyclic Prefix) [10].

As in LTE and 5G, the smallest physical resource is the Resource Element (RE), given by one subcarrier on frequency domain, and one OFDM symbol. In NR, the RE is mapped in frequency-time in a different way, according to the numerology adopted. A NR Resource Block (RB) consists of 12 consecutive subcarriers in the frequency domain. Finally, the Resource Grid (RG) is a representation of the available Resource Elements, considering the available subcarriers and symbols.

NR has been conceived to avoid as much as possible "always on" signals. Consequently, in 5G there is only one "always-on" NR signal, the Synchronization Signal/Physical Broadcast Channel (SS/PBCH), called also "SS Block" (SSB), which includes the Synchronization Signal (SS), the Physical Broadcast Channel (PBCH), and the Physical Broadcast Channel Demodulation Reference Signal (PBCH-DMRS), which works as a reference signal for decoding PBCH. For the specific problem on which this paper is focused, it is worth noting that Vector Signal Analysers typically measure the average power per RE of PBCH-DMRS, averaging the PBCH-DMRS power of all the SSBs in the SSB burst. Regarding the structure of the SSB, an SSB is mapped to four OFDM symbols in the time domain and 240 contiguous subcarriers (20 RBs) in the frequency domain. Furthermore, it is also concentrated in space since the SSB is transmitted using directive beams. Loosely speaking, SSBs are a directional version of synchronization signals that are transmitted with high periodicity, while the CRS in 4G is distributed in the frames, and is transmitted in the whole cellular system sector. This point will be better clarified in a following subsection.

### 3.2. Access Mode Frequency Division Duplexing (FDD) VS. TDD

The access mode implemented by the signal plays a crucial role on the definition of an extrapolation technique. The 5G systems mainly implement the TDD mode, where downlink and uplink transmissions share the same carrier frequency, being separated by a rigid time schedule. On

the other hand, the preferred access mode used by 4G systems is the Frequency Division Duplexing (FDD) (although several TDD configurations are also allowed for 4G systems [11], and extensively studied in [12]), where downlink and uplink transmissions occupy different frequencies. Figure 1 shows a zero span measurement, i.e., the time evolution of the received power at a given frequency, for a 4G (a) and a 5G (b) source.

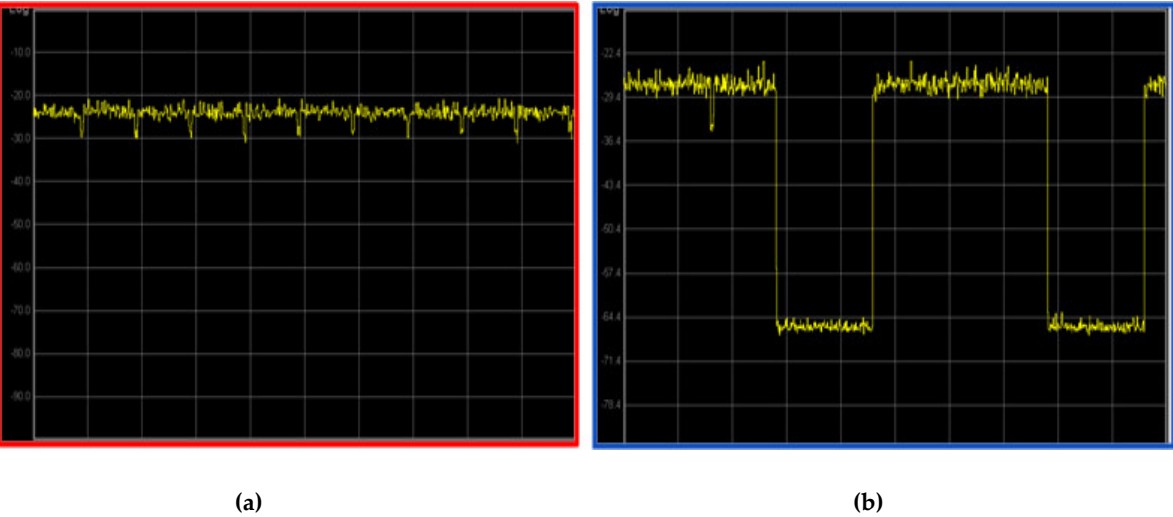

**(a)**                                                        **(b)**

**Figure 1.** Zero span measurement for a fourth generation (4G) Frequency Division Duplexing (FDD) (**a**) and a 5G Time Division Duplexing (TDD) source (**b**). A TDD source does not transmit during the uplink slots, implying an overall reduction of the instant maximum received power.

The exposure to the electromagnetic field generated by a FDD source is quite different when compared to that from a TDD source. For a FDD system, the source is 'always on', i.e., the transmission at the carrier frequency is continuous. A TDD system, instead, alternates downlink and uplink slots, resulting in an intermittent transmission, which implies an overall reduction of the instant maximum received power. As a consequence of the TDD access mode, a duty cycle-related corrective factor—in the following referred to as $F_{TDC}$—must be included in a 5G extrapolation proposal.

### 3.3. Passive VS. Active Antennas: Beam Sweeping

The major improvement related to the switch from a 4G to a 5G network is the use of active antennas. Thanks to these smart devices, the operators can maximize the efficiency of their valuable spectrum resources to increase network capacity. Prior to 5G, antennas were linked to a fixed radiation pattern whose shape and beam direction does not change in time. The 5G systems equipped with active antennas are instead able to synthesize many beams pointing toward different directions, and dynamically reconfigurable according to the electromagnetic environment. This feature modifies dramatically the transmission mode of the 5G broadcast channels compared to 4G systems. As an example, Figure 2 shows a fixed beam as used in 4G base stations (a) and the corresponding zero span measurement of the received power of the CRS (b).

Due to the static nature of the radiation pattern, the CRS received power level is almost constant in time. This is a very attractive characteristic for a pilot signal used as a power reference in an extrapolation method.

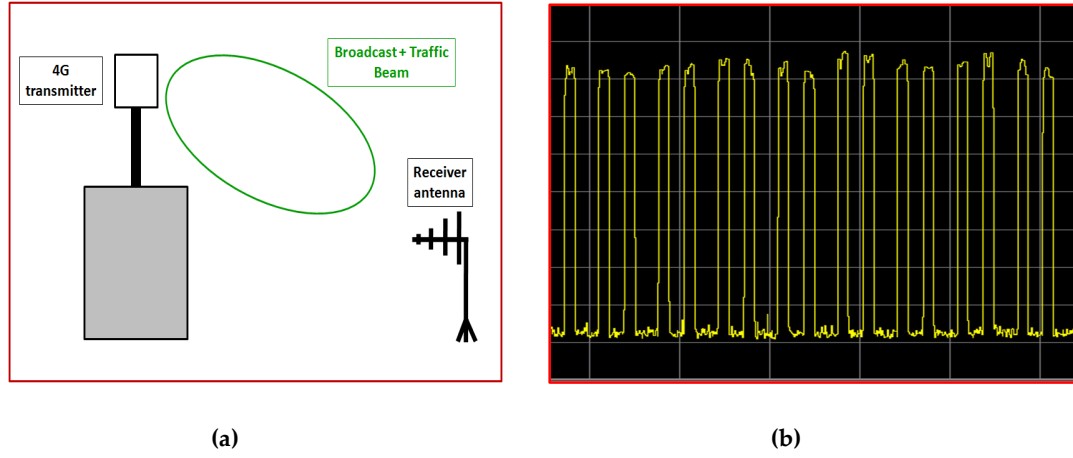

**Figure 2.** Static beam synthesized by a 4G passive antenna (**a**) and zero span measurement of the Cell-Specific Reference Signal (CRS) received power level (**b**). Due to the static nature of the radiation pattern, the CRS received power level is almost constant in time.

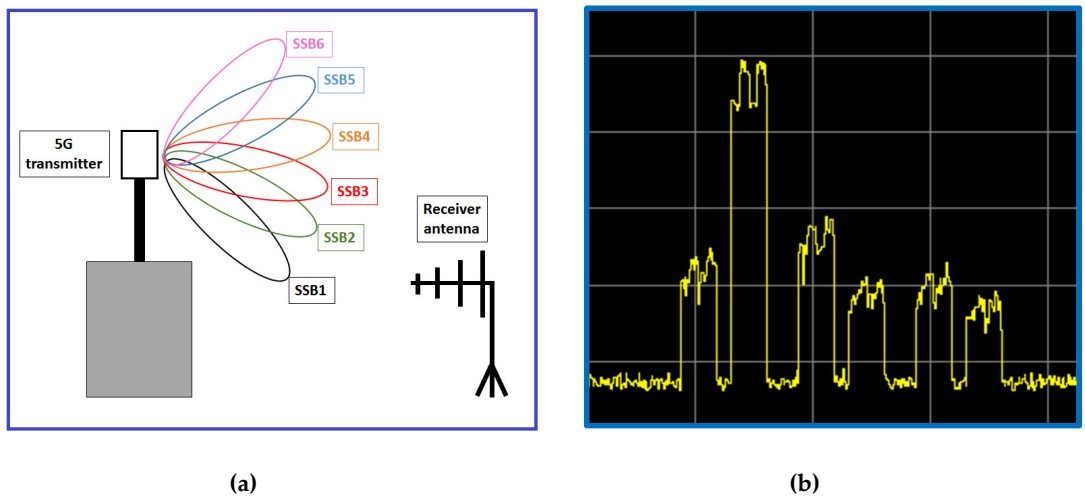

**Figure 3.** Beam sweeping synthesized by a 5G active antenna (**a**) and zero span measurement of the Synchronization Signal Blocks (SSBs) received power level (**b**). The received power is not constant in time, since each SSB is transmitted from a specific beam during a dedicated time slot. A detailed explanation on the SSBs is discussed in [7].

Coming to the last generation of cellular systems, Figure 3 (a) shows the dynamic beam sweeping strategy implemented by 5G systems to transmit the SSBs. Each beam is associated to a SSBs transmission (in the example ranging from SSB1 to SSB6). In Figure 3 (b) the corresponding zero span measurement of the SSBs received power is plotted. The figure shows that, due to the beam sweeping strategy, the received power for the pilot channels packed into a SSB is no longer constant in time. This implies that some serious considerations have to be done in order to identify channel that is more suitable as a power reference in an extrapolation method for 5G signals. In the following, the issue concerning time-varying received SSB power will be addressed by the definition of a proper factor, that will be caller R, based on the SSBs measurement carried out by a Vector Signal Analyser [7]. The use of a Vector Signal Analyser is advantageous since it allows an accurate identification of the SSBs also when data are transmitted in REs located in the same temporal slots used by SSBs, making identification of the SSB level cumbersome using time domain measurements.

*3.4. Passive VS. Active Antennas: Beamforming*

Active antennas used for 5G sources are also able to implement beamforming, i.e., a signal processing technique used for high gain, directional signal transmission. Beamforming is achieved by combining elements in an antenna array in such a way that signals at particular angles experience constructive interference while others experience destructive interference. Due to the beamforming feature of 5G, the transmitting antenna is allowed to synthesize a high gain pattern toward the user equipment to support high data traffic (Figure 4).

This feature has a large impact in the definition of an effective extrapolation technique. For the first time since the appearance of 2G systems, we are obliged to face a change of paradigm for the instant maximum extrapolation. For 5G systems, in fact, the received power associated with the broadcast channels (i.e., the signals packed into SSBs) can be no longer a reference for the maximum receivable power level, since the traffic data can be transmitted by a dedicated pattern characterized by a higher gain compared to the SSB beams. Therefore, the received power of the broadcast channel elected as a reference for the 5G system has to be corrected by a beamforming factor ($F_{beam}$ in the following).

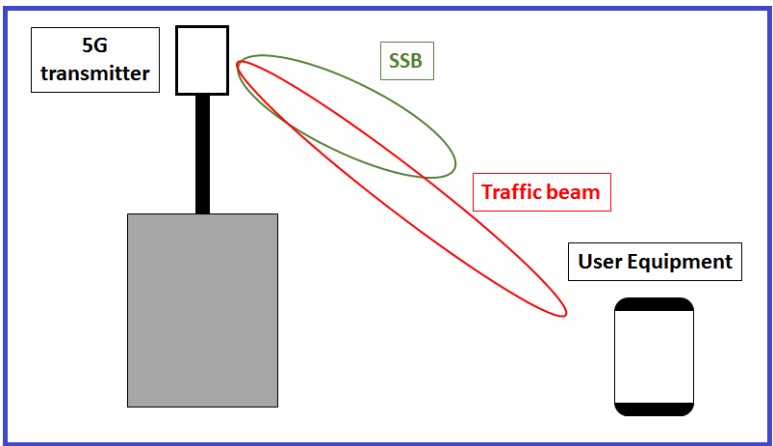

**Figure 4.** Representation of the effect of beamforming. While the SSBs are associated to a finite number of pre-selected beams, data traffic are transmitted using a beam focused toward the user equipment.

As an example, the effect of beamforming can be appreciated with a zero span measurement shown in Figure 5 where both SSBs and traffic data for a 5G signal are presented. The figure shows a $F_{beam}$ value equal to 5 dB.

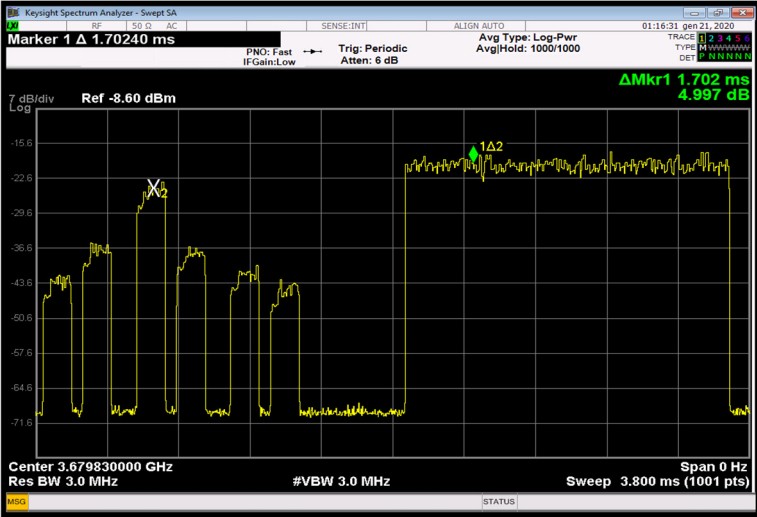

**Figure 5.** Zero span mode measurement; on the left the SS Burst containing 6 SS Blocks; on the right, signal associated to user data; user data are transmitted using a +5 dB higher gain beam.

## 4. Proposed Extrapolation Technique for 5G Signals

The aim of this section is to describe an extrapolation technique for 5G signals developed by the researchers of ARPA Lazio and currently object of an extensive test.

The maximum EMF level in a given location is estimated by the product of three factors

$$E_{5G}^{max} = \sqrt{N_{sc}(B,\mu) \cdot F_{TDC}} \cdot E_{RE}^{max} \tag{4}$$

where:

- $N_{sc}(B,\mu)$ is the total number of subcarriers of the NR carrier, equal to twelve times the total number of Resource Blocks $N_{RB}$ (RBs) available for the signal (we recall that a Resource Block consists of 12 consecutive subcarriers in the frequency domain for a time length that depends on the numerology μ [7,10]).This parameter depends on both the signal bandwidth B and the numerology μ, as specified in Table 5.3.2-1 and 5.3.2-2 of [3];

- $F_{TDC}$ is the deterministic scaling factor representing the duty cycle of the signal, i.e., the fraction of the signal frame reserved for downlink transmission (Section 3.2);

- $E_{RE}^{max}$ represents the maximum EMF level measured for a single Resource Element (RE), i.e. the smallest unit of the resource grid made up of one subcarrier in frequency domain and one OFDM symbol in time domain.

It is evident that the definition of $E_{RE}^{max}$ plays a crucial role in Equation 4. As already discussed in the previous sections, the major issues are concerned with the choice of the pilot channel whose received power is eligible to be the reference for the extrapolation technique. In addition, the effect of some peculiar 5G characteristics that influences the received power of the pilot channel, such as beam sweeping and beam forming produced by the usage of MU-MIMO antennas, has to be taken into proper account.

In order to harmonize 4G and 5G extrapolation techniques, the proposed pilot channel in this paper is the PBCH Demodulation Reference Signal (PBCH-DMRS). As discussed in Section 3.1, the PBCH-DMRS is a component of the SSBs, and its physical location is determined by the Physical Cell ID, as reported in Table 7.4.3.1-1 of [13]. According to this choice, the maximum EMF level for a single RE can be defined as

$$E_{RE}^{max} = \langle E_{RE}^{PBCH-DMRS} \rangle \sqrt{\frac{F_{beam}}{R}} \tag{5}$$

where:

- $\langle E_{RS}^{PBCH-DMRS} \rangle$ is the average received EMF level for PBCH-DMRS for a single RE;

- $R$ is defined as the ratio of the average detected power of all the SSBs in a burst to the power of the stronger SSB in the burst [7]. This parameter accounts for the effect of the beam sweeping (Section 3.2 and Figure 3) on the received EMF level of all the SSB in a burst, allowing for a precise estimate of the maximum received EMF level for PBCH-DMRS for a single RE, starting from the direct evaluation of $\langle E_{RE}^{PBCH-DMRS} \rangle$;

- $F_{beam}$ is a parameter, which takes into account the effect of a potential boost of the traffic beams with respect to maximum EMF level received from the pilot channel, due to the effect of beamforming produced by the usage of MU-MIMO antennas (Section 3.4 and Figures 4 and 5).

It worth noting that the choice of $\langle E_{RE}^{PBCH-DMRS} \rangle$ as a reference EMF level for a single RE is particularly effective since it is directly measurable using a VSA. It represents the average received EMF level of the PBCH-DMRS channel from all the SSBs in a burst, measured on a RE basis. As a consequence of the beam sweeping, in fact, the received EMF level is different for each SSB, according to the relative orientation between the SSB beam and the receiver antenna (Figure 3).

Finally, the $F_{beam}$ parameter is related to the beamforming feature of 5G that allows to synthesize a high gain pattern toward the user equipment to support high data traffic (Figure 4). The parameter $F_{beam}$ represents the EMF level boost of a RE transmitted by the traffic beam with respect to the received EMF level corresponding to the DMRS-PBCH channel of the most intense SSB.

## 5. Validation of the Extrapolation Technique

The extrapolation technique for quantifying the instant maximum EMF level received from a 5G source proposed in the previous section has been validated by several experimental measurements on a set of 5G signals. In order to have realistic signals in fully controlled conditions, signals transmitted by commercial FR1 5G base stations, measured OTA (Over the Air) [7], were replicated in laboratory.

The precise replication and measurement of the signals is obtained using the experimental set-up shown in Figure 6. The set-up consists of a Keysight Technologies N5172B EXG X-Series Vector Signal Generator and a Keysight Technologies N9020A MXA Vector Signal Analyzer [14] connected by a coaxial cable. Both the instruments are equipped with a dedicated software for both the generation of the transmitted signal with full control of all the signal parameters, as well as an accurate analysis of the received 5G signals in time domain (span-zero analysis), frequency domain, and code domain.

In order to investigate the role of each of the correction factors included in Equations 4 and 5 (i.e., $F_{TDC}$, $R$ and $F_{beam}$), three different 5G signals of increasing "complexity" from the point of view of application of the extrapolation formula are considered. As noted above, the structure of the signals matches the characteristics of an OTA signal in the FR1 (sub-6 GHz) band transmitted by a commercial 5G base station [7]. The characteristics of the 5G signals generated for this study are listed in Table 1.

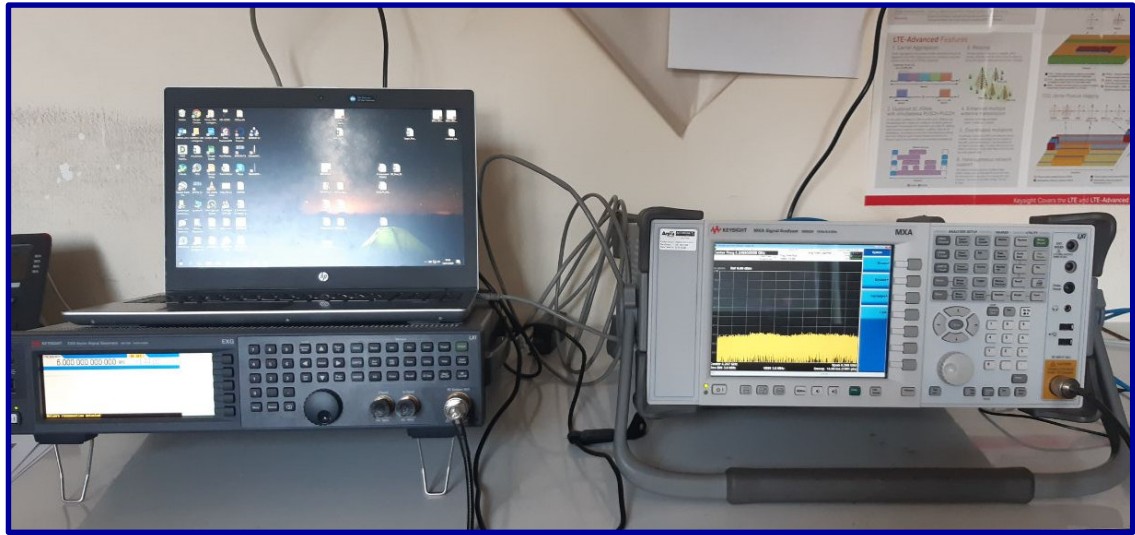

**Figure 6.** The experimental setup. The generated 5G signals are carried by a coaxial N-type cable from a Vector Signal Generator (on the left) to a Vector Signal Analyzer (on the right). A personal computer (PC) running a demodulation 5G software is also used.

### 5.1. FDD 5G Signal (Signal #1, $\langle E_{RE}^{DMRS-PBCH} \rangle$ Measurement Procedure)

As first example, we consider a FDD signal, indicated as Signal #1, whose parameters are listed in the first column of Table 1.

Since Equation 4 is focused on the determination of the instant maximum EMF level produced by a 5G source, the sample signals are generated in an ideal condition of full traffic, i.e., the entire signal bandwidth is devoted to downlink transmission. The resource allocation of the signal is shown in Figure 7. The figures shows a SS Burst, containing six SS-PBCH (drawn in yellow in the figure), and the DL-SCH (DownLink BroadCast Channel, in green). Since DL-SCH encodes user data and

paging information, in condition of full traffic and FDD duplexing it fills all the available resources. The encoded data is then mapped to a physical downlink shared channel (PDSCH). A screenshot of the output of the VSA 5G software is shown in Figure 8, showing some characteristics of the signal.

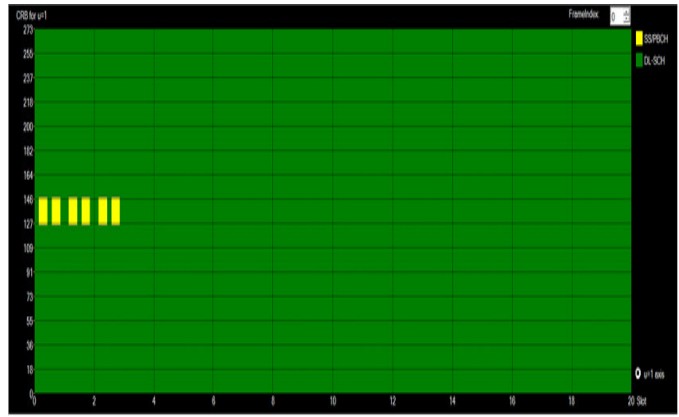

**Figure 7.** FDD signal #1 (FDD) frame structure. Green blocks represent the resources allocated to data transmission.

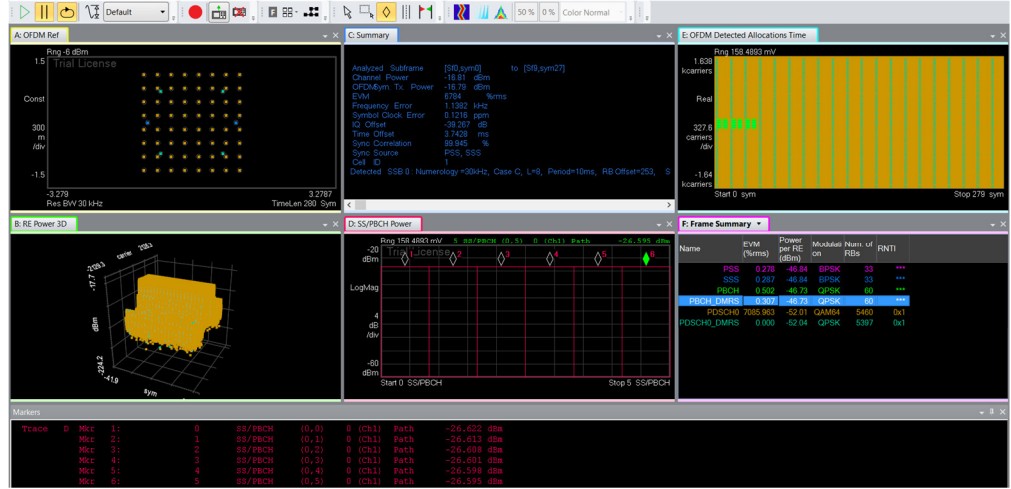

**Figure 8.** Screenshot of the 5G demodulation software with many relevant characteristics of the detected signal #1, such as the In-phase and Quadrature (IQ)diagram, the carrier vs. time allocation map, and the detected power level for control channels, data channels, and SS-Blocks.

| Name | EVM (%rms) | Power per RE (dBm) | Modulation | Num. of RBs | RNTI |
|---|---|---|---|---|---|
| PSS | 0.278 | -46.84 | BPSK | 33 | *** |
| SSS | 0.287 | -46.84 | BPSK | 33 | *** |
| PBCH | 0.502 | -46.73 | QPSK | 60 | *** |
| PBCH_DMRS | 0.307 | -46.73 | QPSK | 60 | *** |
| PDSCH0 | 7085.963 | -52.01 | QAM64 | 5460 | 0x1 |
| PDSCH0_DMRS | 0.000 | -52.04 | QPSK | 5397 | 0x1 |

**Figure 9.** Frame summary with the average received power for the PBCH-DMRS channel.

As discussed in Section 4, the choice of the PBCH-DMRS as measured parameter for the extrapolation procedure is particularly attractive since this is one of the parameters directly available on the VSA. In Figure 9, an example of frame summary given by the VSA is shown. PBCH-DMRS

average power is one of the standard parameters evaluated by the 5G demodulation software of the VSA. The measured value of the correction factors to be used in Equations 4 and 5 for signal #1 is listed in the first row of Table 2. The $\langle E_{RE}^{DMRS-PBCH}\rangle$ and $E_{5G}^{max}$ value estimated from the correction factors are reported in the first two rows of Table 3.

In order to validate the extrapolation method, the $E_{5G}^{max}$ is compared with the result of a channel power measurement, which is an accurate estimate of the total power instantaneously carried by the signal. Note that this comparison represents a good validation method since the instantaneous EMF level from a channel power measurement corresponds to the maximum EMF level, under the assumption of a 100% resource allocation for data transmission.

The result of the channel power measurement, reported in the last row of Table 3, is in excellent agreement with the extrapolated data.

### 5.2. TDD 5G Signal (Signal #2, $F_{TDC}$ Measurement Procedure)

The above-described validation regards a FDD signal, and consequently does not involve the parameter that is associated to the duty cycle of the signal.

As a second example a TDD signal, indicated as Signal #2 and whose parameters are listed in Table 1, is considered. Signal #2 has the same parameters of the OTA measured signal described in Section 3.2 of [7] but the amplitude of the SSB does not change. Consequently, the evaluation of the $F_{TDC}$ parameter and $\langle E_{RE}^{DMRS-PBCH}\rangle$ value, but not of *R*, is required for the extrapolation technique. According to the specific TDD schema implemented by the signal, about 74% for signals #2 is allocated for downlink data transmission [7].

Figure 10 shows the allocation of the resources in case of TDD signal. Moreover, in case of Signal #2, the signal is generated in an ideal condition of full traffic. However, due to the TDD duplexing, part of the resources are reserved for the uplink (drawn as grey pixels in Fig. 10) according to the specific TDD scheme adopted in the transmission.

Estimation of $\langle E_{RE}^{DMRS-PBCH}\rangle$ is obtained directly from VSA 5G software output following the procedure described in Section 5.1. The measured value of $\langle E_{RE}^{DMRS-PBCH}\rangle$ is listed in the first row of Table 3.

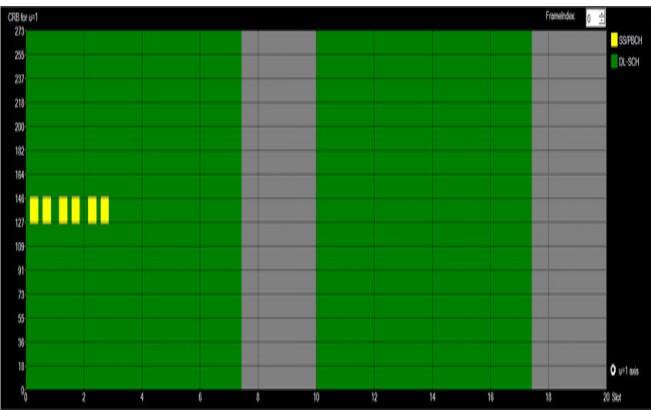

**Figure 10.** TDD signals #2 frame structure; green and grey blocks represent the resources allocated to data transmission and those allocated to the uplink transmission, respectively.

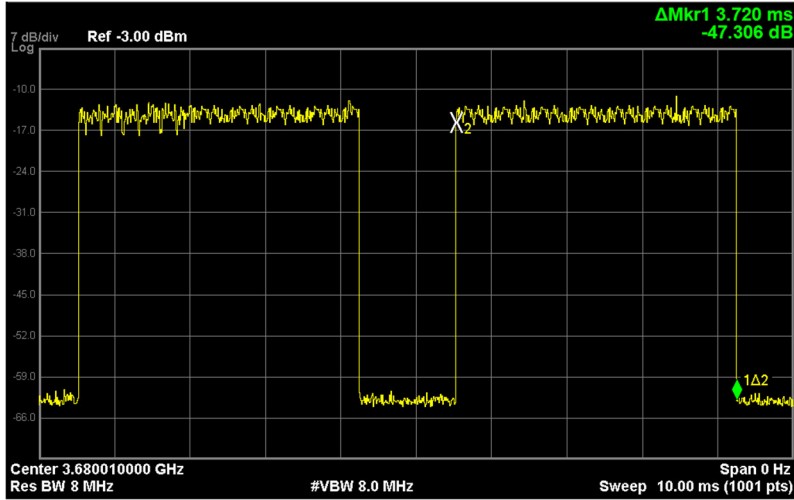

**Figure 11.** Zero span measurement of signal #2 for a duty-cycle related corrective factor ($F_{TDC}$) determination.

The estimation procedure of $F_{TDC}$ is described in [7]. For sake of reader convenience, in the following the procedure is briefly described. The estimation of F$_{TDC}$ factor is based on a zero span measurement using the following settings:

1.  center frequency = the same of the SS-Blocks;
2.  resolution bandwidth = the entire signal bandwidth (or in the case of hardware limitations, the largest resolution bandwidth (RBW) allowed by the instrument);
3.  sweep time = a multiple of the frame length (10 ms);
4.  periodic trigger = 10 ms;
5.  detector trace = max-hold;
6.  acquisition time = 10 sec (or the lowest time needed by the downlink slots to properly arise).

Figure 11 shows the zero span spectrum with the measurement of the duration of the downlink transmission $T_{dw} = 3.720\ ms$. The measured value for $F_{TDC}$ is then obtained as the ratio of $T_{dw}$ to the period of TDD scheme (5 ms, for signal #2), resulting in $F_{TDC} = 0.744$.

The measured value of the correction factors to be used in Equations 4 and 5 for signal #2 is listed in the first row of Table 2. The $E_{5G}^{max}$ values estimated from the correction factors is reported in the second two rows of Table 3. A comparison of and $E_{5G}^{max}$ value with the result of channel power measurement (third row of Table 3) shows again an excellent agreement.

### 5.3. TDD 5G Signal (Signal #3, R, $F_{beam}$ Measurement Procedure)

Signal #2 has a constant SSB level in the SS Burst. As discussed in the Sections 4, beam sweeping technique causes a variation of the received SSB level. Signal #3 includes this effect. The parameters of the signal #3 are the same of the OTA measured signal discussed in Section 3.2 of [7], and are listed in the third row of Table 1. The effect of beam sweeping on the SSB level is simulated by a different power boost to each generated SSB. It is worth stressing that the power boost accurately simulates a real, on-field SSB measurement of a signal in FR1 band [7], and that the detected power level for the SSBs belonging to the same burst is not constant, due to the time-scheduled beam management (cf. Figure 3). Finally, the effect of the beamforming (see Figure 5) is simulated by applying a 5 dB boost to the EMF level of a RE transmitted by the traffic beam with respect to the EMF level corresponding to the DMRS-PBCH channel of the most intense SSB.

In this case, extrapolation technique requires the estimation of all the parameters: $\langle E_{RE}^{DMRS-PBCH} \rangle$, $F_{TDC}$, $R$ and $F_{beam}$. The procedure for the estimation of the first two parameters has been described in Section 5.1 and 5.2, and will not be repeated. Instead, we will focus our attention toward the estimation of the $R$ parameter. The procedure is described in [7], wherein the parameter $R$ has been introduced to evaluate the effect of beam sweeping on the SS-Block detected power level as:

$$R = \frac{<P_{SS-Block}>}{P_{SS-Block}|_{max}} \tag{6}$$

where <P$_{SS-Block}$> is the average detected power of all the SS-Blocks in a burst and P$_{SS-Block|max}$ is the power of the strongest one.

The SS Burst contains six SS Blocks [7], but only four of them are detected as above the noise level. The received power level of the detected SSBs is shown in Figure 12.

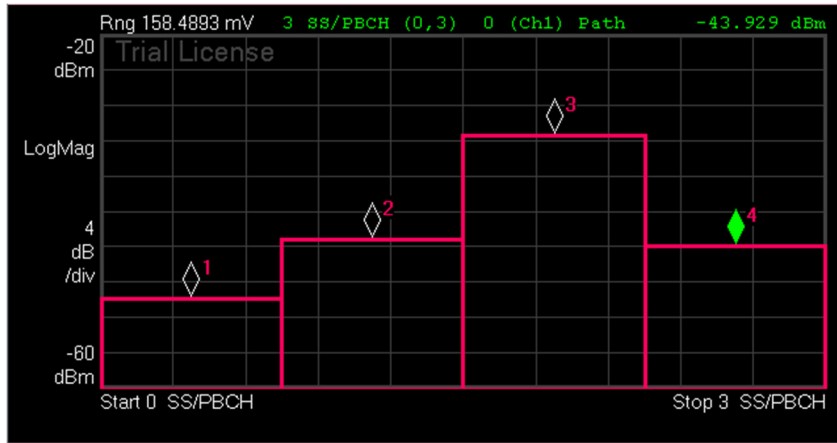

**Figure 12.** Received power level of SSBs for signal #3 where the effect of beam sweeping is simulated; due to a strong power de-boost, only four SSBs are detected.

The demodulation software provides a measurement of the received power level of all the SSBs, allowing an accurate estimation of the beam sweeping R factor.

Finally, $F_{beam}$ is estimated using zero-span mode and measuring the difference between the EMF traffic beam level and EMF level corresponding to the DMRS-PBCH channel of the most intense SSB, obtaining a value of $F_{beam} = 3.162$.

A comparison of and $E_{5G}^{max}$ value with the result of channel power measurement (Table 3, signal #3 case) shows an excellent agreement also in this case.

## 6. Conclusions

In this paper, an extrapolation procedure for 5G signals is proposed. The procedure includes the complex effects of the signals radiated by the 5G cellular base stations currently in use or in deployment by means of four parameters: $F_{TDC}$, $\langle E_{RE}^{DMRS-PBCH} \rangle$, R, and the $F_{beam}$. Procedures for estimation of these parameters using a VSA are presented. In particular, $F_{TDC}$ and $F_{beam}$ are measured using span-zero mode, while $\langle E_{RE}^{DMRS-PBCH} \rangle$ and $R$ are measured in the resource code domain. We must note that estimation of the $F_{beam}$ factor in OTA measurements requires to force data link toward the measurement point. A possible solution is to establish a connection toward the base station using a proper receiver close to the measurement point forcing a full-loaded traffic. This point is currently object of research.

In general, the studies carried out in controlled conditions using signals that are copies of OTA measured signals in FR1 band show an excellent agreement between channel power measurements and the extrapolated values in all the cases under test. All the results obtained up to the writing of this paper suggest that the proposed extrapolation method is an effective tool for a correct evaluation of the instant maximum EMF level radiated by a 5G sources. OTA measurements are scheduled to complete the validation of the technique in real conditions.

**Table 1.** Characteristics of the 5G signals.

|  | Signal #1 | Signal #2 | Signal #3 |
|---|---|---|---|
| Center frequency [MHz] | 3680.01 | 3680.01 | 3680.01 |
| Bandwidth [MHz] | 100 | 100 | 100 |
| Numerology [μ] | 1 | 1 | 1 |
| SS-Block center frequency [MHz] | 3679.83 | 3679.83 | 3679.83 |
| # of SS-Block per SS-Burst | 6 | 6 | 6 |
| Access Mode | FDD | TDD | TDD |
| TDD scheme | - | DDDDDDDSUU | DDDDDDDSUU |
| Power boost for SSBs | - | - | Yes |
| Power boost for traffic channel | - | - | 5 dB |
| Data scenario | Full traffic | Full traffic | Full traffic |

**Table 2.** Correction factors for the 5G signals investigated.

|  | Signal #1 | Signal #2 | Signal #3 |
|---|---|---|---|
| $N_{sc}$ | 3300 | 3300 | 3300 |
| $F_{TDC}$ | 1 | 0.744 | 0.746 |
| $R$ | 1 | 1 | 0.285 |
| $F_{beam}$ | 1 | 1 | 3.162 |

**Table 3.** Comparison between the channel power and the instant maximum EMF level extrapolated by Equations 4 and 5.

|  | Signal #1 | Signal #2 | Signal #3 |
|---|---|---|---|
| $\langle E_{RE}^{DMRS-PBCH} \rangle [\frac{V}{m}]$ | $1.030 \times 10^{-3}$ | $1.021 \times 10^{-3}$ | $3.133 \times 10^{-4}$ |
| $E_{5G}^{max} [\frac{V}{m}]$ | $5.927 \times 10^{-2}$ | $5.069 \times 10^{-2}$ | $5.183 \times 10^{-2}$ |
| $Channel\ Power [\frac{V}{m}]$ | $5.929 \times 10^{-2}$ | $5.093 \times 10^{-2}$ | $5.152 \times 10^{-2}$ |

**Author Contributions:** Conceptualization, D.F., S.C., E.G., and S.P.; Formal analysis, D.F.; Methodology, D.F., E.G., and S.P.; Supervision, T.A., R.C., M.D.M.; Writing—original draft, D.F., M.D.M. All authors have read and agreed to the published version of the manuscript.

**Funding:** This research was partially funded by the program 'Dipartimenti di Eccellenza (2018–2022)' of MIUR: Italy.

**Conflicts of Interest:** The authors declare no conflict of interest.

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
