# Peer review of "Experimental Procedure for Fifth Generation (5G) Electromagnetic Field (EMF) Measurement and Maximum Power Extrapolation for Human Exposure Assessment"

_environments, doi:10.3390/environments7030022_

Round 1

Reviewer 1 Report

It seems that the authors revised the original version of their paper and published it elsewhere (see ref. 6: D. Franci, S. Coltellacci, E. Grillo, S. Pavoncello, T. Aureli, R. Cintoli, M. D. Migliore, "An Experimental Investigation on the Impact of Duplexing and Beamforming Tecnhiques in Field Measurements of 5G Signals", Electronics 2020, 9(2), 223).

Concerning the current manuscript under review, which is actually a new one, the authors should make clear and underline which is the novelty of their work, especially compared with their other paper mentioned above (ref. 6), since it seems that the current manuscript is heavily based on ref. 6.

The authors should describe and justify better the choice of the proposed pilot channel for their method, which is the PBCH Demodulation Reference Signal (PBCH-DMRS). This should be done also in comparison with other possible choices.

The authors state that “in order to validate the extrapolation method, the ?????? is compared with the result of a channel power measurement… this comparison represents a good validation method since the instantaneous EMF level from a channel power measurement corresponds to the maximum EMF level, under the assumption of a 100% resource allocation for data transmission”. The authors should further describe the performed channel power measurement method in the presented cases and justify its validity and purpose.

The authors should explain all abbreviations used at their first occurrence (e.g. MU MIMO in line 37, or correct the existing ones, e.g. SDMA in line 154).

The last reference should be ref. 11.  

The text has some serious grammatical/syntax errors and needs a thorough linguistic revision (e.g. in lines 18, 43, 151, 220, 237)

Reviewer 2 Report

The authors have improved the manuscript, but there are still som major parts that needs attention. I still miss general technical information about the 5G system, this is a new system and not very many readers have insight in this. Why have the three 5G signals been chosen? Is it correct in Table 1 that a Sub-6 GHz signal was included, the center frequency is specified at 3.68 GHz? The lab signal is FDD while the other are TDD why? Why didn´t you try to replicate the commercial signal in the lab with the same technical characteristics? You write that the aim is to form a starting point for further investigation of the power of 5G signals, what are the challenges for 5G signals compared to other systems (UMTS, GSM etc.) that needs to be solved before a measurement methods can be established? And after this investigation, what is left before a method could be implemented, this should be discussed in the paper (discussion). Since the aim vague it is difficult to understand what is the actual results in this study, By improving the aim, for instance by adding specific aims that are related to the results the impact of the paper will be clearer. Figure 3 is barely mentioned as a result and not discussed. You could go into more detailed explaining what is seen in Figure 9, for instance the SS-Blocks. The discussion needs to be heavily improved, since it is not obvious what is the actual result of the study.

The paper has been rewritten and shortened to only include maximum power extrapolation. Meanwhile the authors have published another paper in Electronics 2020, 9, 23 (ref no 6 in present paper) that includes the major part of the previous submitted paper and there are similarities between the published paper and the submitted paper that can’t be ignored.
However, the authors have focused the submitted paper and have a clear aim, but with respect to the publication above and also that the paper still suffers from more looking like a technical report and the fact that this is the third round with sparse improvements on that, I would recommend to reject this paper.

Round 2

Reviewer 1 Report

Since the manuscript is heavily based on ref. 10 and it reports only preliminary experimental investigation results carried out in a laboratory environment, i would not suggest its publication as a research paper at the present time. It could be reconsidered for publication as a research paper, only after OTA measurements are performed in order to ensure the validation of the proposed method in real conditions.

Author Response

Dear reviewer,

in the review of the paper please consider that the implementation of 5G is still at the first stage, and the few active base stations are mainly used for tests that allow very limited access to the on-air signals. Furthermore we need stable and repeatable conditions in order to verify the methods. In order to solve these problems, in a first phase a set of signals transmitted during a test of a 5G base station were measured on-air. Then a laboratory set-up was created capable of reproducing the acquired signals. This approach has several advantages. In particular it allows to work with an exact copy of real 5G signals in laboratory, whose parameters are completely known. Furthermore, it is possible to vary some signal parameters to consider different cases. For example, in the experimental examples reported in this article the proposed extrapolation technique is applied to signals having an increasing ‘degree of complexity’."

The extrapolation technique procedure as described in this paper is complete. Its application in real cases has been so far prevented by the lack of availability of 5G systems working in standard data load conditions.

This manuscript is a resubmission of an earlier submission. The following is a list of the peer review reports and author responses from that submission.

Round 1

Reviewer 1 Report

In this paper, a method for the evaluation of the human exposure to the electromagnetic fields produced by 5G radio base stations, is presented. A strength of this paper is that the proposed measurement methods can be used by the regulatory authorities for assessing compliance with the national exposure limits.

In order for this paper to be published, the authors should pay attention to the following points:

Since the authors claim that their work could be used by the regulatory authorities for assessing compliance with the general public EMF exposure limits, such a specific example with actual data should be added using the presented results for a base station in Figure 9 and equations 3-4. The authors should also comment and quantify the overestimation that their method could induce and define which data is needed by the network operators in order to acquire results for assessing compliance with the exposure limits.

Specific points need further definitions and/or explanatory text in order to be better understood by the reader. In this way, the text would be easily read without the need to consult the relevant references first, as R in lines 137,188 and βTDD in line 153.

Certain key references are missing, such as for the equipment used (lines 47-48, 50), equations 2 (in line 165), 3 (in line 204) and 4 (in line 209).

The IEC TR 62669:2019 should also be referenced and taken into account.

Some minor grammatical/syntax errors exist, e.g. in lines 52, 129.

Reviewer 2 Report

This paper gives an good insight in the 5G technology which is necessary yo have in order to do accurate exposure assessments. The authors have included a test signal as well as two commercially available 5G systems. The results are of interest and worth publishing but the manuscript needs several improvements to enhance the impact and understanding.

The paper includes many technical terms and abbreviations that might not be clear to the reader of Environments for instance: MIMO, TDD, beem sweeping are not defined/explained.

The introduction in sparse, some text with a general explanation of the 5G system, including technique would be of interest to the reader. Also to understand that the commercial 5G system in this article are only some examples of the whole 5G spectrum.

The materials and Methods are also very sparse. Please write in text how the measurement procedure was done, not just a list of equipment.

The paragraphs in the results section containing info on the mobile systems used should be moved to the material and methods.

Figure 6 lacks information on the x and y axes.

To me, the twelve SS blocks are not easy to recognize in Figure 7, line 127. More information on beem sweeping would also be of interest.

How representative are the power levels in table 1? What are the max and mean, how many available power levels are the 5G system offer?

The discussion are vague  and the results of the study is barely mentioned. Again, the representativeness of the measured results should be discussed, possible errors, but also how the results can be used in future investigations

The conclusion is missing.

Reviewer 3 Report

Section I:

The state of the art SOTA is completely missing, please consider several IEC standards (also 5G), FCC, CENELEC, all relevant publications, etc.

Explain all abbreviations at first occurrence

Explain novelty compared to SOTA

Section II: this is not a method section: only some bullets about equipment. No method, no motivation, no settings, no discussion, are the authors sure they submitted the right version. This looks like a measurement report?

Section III:

In general: all discussion, motivations are missing: probably this is just a wrong version of the manuscript.

Fig. 1-2 are screen shots from a spectrum analyzer measurement, Fig. 4-7 too. Explain these at least. You can discard some.

Fig. 2: is a very “clean” lab example, in practice very different. Maybe vary for different test signals with different SCS, BW, DL and UL user data etc.

8: you mention “strong evidence”: explain, validation? What is the deviation and what is the ground truth: this is not explained.

9: next step is the determination of the total power: so the method is not complete. First complete the method.

Conclusions are missing.

Round 2

Reviewer 2 Report

The authors have improved the manuscript, but there are still som major parts that needs attention.

I still miss general technical information about the 5G system, this is a new system and not very many readers have insight in this.

Why have the three 5G signals been chosen? Is it correct in Table 1 that a Sub-6 GHz signal was included, the center frequency is specified at 3.68 GHz?

The lab signal is FDD while the other are TDD why? Why didn´t you try to replicate the commercial signal in the lab with the same technical characteristics?

You write that the aim is to form a starting point for further investigation of the power of 5G signals, what are the challenges for 5G signals compared to other systems (UMTS, GSM etc.) that needs to be solved before a measurement methods can be established? And after this investigation, what is left before a method could be implemented, this should be discussed in the paper (discussion).

Since the aim vague it is difficult to understand what is the actual results in this study, By improving the aim, for instance by adding specific aims that are related to the results the impact of the paper will be clearer.

Figure 3 is barely mentioned as a result and not discussed.

You could go into more detailed explaining what is seen in Figure 9, for instance the SS-Blocks.

The discussion needs to be heavily improved, since it is not obvious what is the actual result of the study.

Reviewer 3 Report

see previous comments: this is more a measurement report not a paper, novelty compared to IEC, Keller and references provided is not enough

work on a more extensive study, this paper was rejected but mdpi stragely provided a review option